# Root Cause Analysis of Failures in Microservices through Causal Discovery

**Azam Ikram**[1]     **Sarthak Chakraborty**[2]     **Subrata Mitra**[2]     **Shiv Kumar Saini**[2]
**Saurabh Bagchi**[1]     **Murat Kocaoglu**[1]
[1]Purdue University, USA     [2]Adobe Research, India
{mikram,sbagchi,mkocaoglu}@purdue.edu
{sarchakr,sumitra,shsaini}@adobe.com

## Abstract

Most cloud applications use a large number of smaller sub-components (called microservices) that interact with each other in the form of a complex graph to provide the overall functionality to the user. While the modularity of the microservice architecture is beneficial for rapid software development, maintaining and debugging such a system quickly in cases of failure is challenging. We propose a scalable algorithm for rapidly detecting the root cause of failures in complex microservice architectures. The key ideas behind our novel hierarchical and localized learning approach are: (1) to treat the failure as an intervention on the root cause to quickly detect it, (2) only learn the portion of the causal graph related to the root cause, thus avoiding a large number of costly conditional independence tests, and (3) hierarchically explore the graph. The proposed technique is highly scalable and produces useful insights about the root cause, while the use of traditional techniques becomes infeasible due to high computation time. Our solution is application agnostic and relies only on the data collected for diagnosis. For the evaluation, we compare the proposed solution with a modified version of the PC algorithm and the state-of-the-art for root cause analysis. The results show a considerable improvement in top-$k$ recall while significantly reducing the execution time.

## 1   Introduction

Root Cause Analysis (RCA) is widely used to ensure the reliability of production systems in many domains such as IT operations [25], telecommunications [35], and medicine [21]. Modern production systems follow a network topology that consists of multiple components connected through complex dependencies. For instance, cloud applications typically follow microservice-based architecture [26]. Such applications have the benefit of a simple development process, uncomplicated maintenance, and flexible deployment [18]. A typical cloud-based application uses several microservices, each performing some small tasks [12]. Each microservice is further instrumented to monitor its state and availability which results in thousands of metrics to monitor [5]. The larger number of metrics makes it challenging to find the root cause of the failures in a timely fashion. It has been reported that it takes on average 3 hours to identify the root cause of a failure without using any automated tools [30]. Since enterprise cloud services providers have prices tied with availability guarantees, a delay in detecting the root cause of a fault can result in significant revenue loss.

Causal structure discovery-based Root Cause Analysis techniques have been used recently to find the root cause(s) of a fault in cloud applications [22, 30, 11, 27, 1, 6, 10]. The goal is to construct a graph with nodes as metrics and a directed edge between two nodes showing the direction and magnitude of the causal effect. A direct application of causal discovery algorithms like PC [27] is infeasible for a microservice system due to a large number of metrics. Existing approaches reduce the number of

36th Conference on Neural Information Processing Systems (NeurIPS 2022).

nodes by using a feature selection approach or selecting a specific set of metrics [28, 13, 30]. An obvious problem with this approach is that the selected set of metrics might not include the root cause metric(s). In addition, the feature selection step might introduce latent variables, which renders the popularly used PC algorithm inapplicable. Furthermore, most of these approaches only rely on observational data and therefore do not utilize the potential invariance present in the interventional data to learn the underlying causal structure.

We make two crucial observations. First, we observe that a fault changes the generative mechanism of the failing node. Hence, a fault can be modeled as an intervention on the failing node, and the data during the fault period as an interventional dataset. We borrow ideas from recent work on learning causal graphs using an interventional distribution with unknown interventional targets [8, 17]. In summary, a binary indicator variable for normal and fault periods is introduced as an additional node. The target metrics of the newly introduced node are the root causes. Second, given the first observation, we do not need to learn the full graph for RCA. Using these two observations, we propose a novel hierarchical and localized causal discovery algorithm, which we call ***Root Cause Discovery (RCD)***, to detect the root cause of a failure. The algorithm pinpoints the root causes without learning the causal structure of the complete graph which allows the algorithm to run on a much larger set of metrics. We further optimize the algorithm by hierarchically exploring the data. We validate RCD on synthetic datasets and data from the Sock-shop application test bed. The results show a significant improvement in runtime and accuracy as compared to the baselines. In addition, we also apply the algorithm to three real-life failures of microservices from a large cloud service provider. RCD correctly identifies the root cause metrics in a fraction of the time than the baseline. Summary of our contributions are as follows:

- We consider failures as interventions on the root cause nodes. Doing so allows us to use distributional invariances not only within the observational, but also across observational and interventional data for learning.

- We propose a novel solution to identify the root causes of the failure using a localized hierarchical learning algorithm. Our algorithm is not only more computationally efficient, but it also requires much fewer anomalous samples compared to the existing approaches.

- We evaluate our system on synthetic datasets and data generated from a real-world application and also report our findings from a production-ready microservice-based application.

Our source code is available online at github.com/azamikram/rcd.

## 2 Background and Motivation

In this section, we provide the background of concepts that underlie our design.

**Causality and Causal graphs.** A variable (metric) is said to cause another variable if a change in the former induces a change in the distribution of the latter. A causal graph is used to encapsulate the causal relationships between variables using a directed acyclic graph (DAG). Every node of the DAG represents a variable, while a directed edge shows the causal relationship between the two variables: $X \rightarrow Y$ indicates that $X$ causes $Y$.

Structural causal models (SCMs) are typically used to model causality between a set of random variables. Accordingly, each variable $X$ is assigned a value based on a functional relation with a subset $Pa_X$ of observed variables and an exogenous noise $E_X$ as $X = f_X(Pa_X, E_X)$. One can then construct the causal graph by assigning the set of observed variables $Pa_X$ as the parents of $X$ for all $X$. Causal Bayesian networks (CBNs) similarly can be used for defining a causal model without specifying the functional relations, and instead specifying observational and interventional distributions through the so-called truncated factorization formula under the causal graph. For a formal treatment of the subject, please refer to [20].

**Intervention.** Intervention on a variable is the process of changing the generative mechanism of that variable. The most widely used notion of interventions are in randomized controlled trials (RCTs), and A/B tests, where we would like to understand the causal effect of a certain treatment. This type of intervention is captured by the do-operator of Pearl [20] and destructs the previous generative mechanism. For example, $do(X = x)$ forces $X$ to take the value of $x$, effectively removing the role of its structural equation. This is also called a *hard intervention*, and its effect in the causal

graph is to sever the incoming edges to the intervened node. On the other hand, *soft interventions* are used to model experiments that do not completely destruct the causal mechanisms but modify them. For example, a soft intervention on $X$ can replace the structure equation $f_X(Pa_X, E_X)$, with some $f'_X(Pa_X, E_X)$, where $f' \neq f$, which then retains the original causal graph but changes the generative mechanism and the conditional distribution i.e., $p(X|Pa_x, do(X = x)) \neq p(X|Pa_X)$. This is known as the faithfulness assumption [20].

**Causal Discovery.** Given a set of observations of random variables from a system, the goal of causal discovery algorithms is to find the underlying causal structure. Most constrain-based causal discovery algorithms estimate the causal graph in two phases. The first phase focuses on building a skeleton using conditional independence (CI) tests, whereas the second part estimates the orientation of the edges between the nodes based on a set of static rules. For the unfamiliar reader, we give a detailed background on the causal graph and causal discovery in the appendix.

Most causal discovery algorithms estimate the causal graph from observational data [19, 20]. However, recently there has been some work done in learning the causal graph from both observational and interventional data as it is more informative than just relying on the observational data [9, 20, 32]. Furthermore, there are a set of studies that tries to find the intervention target (node where the intervention occurred) from the observational and interventional data [8, 29].

**RCA and Causal Inference.** The above mentioned concepts from causal inference can be mapped to the problem of finding the root cause of a failure. Consider that a failure happens at time $t$ on service $X$, then we make the following simplifications;

- At time $t$, a soft intervention is performed on $X$;

- The metric data collected from all the services till time $t$ is an observational (normal) dataset, denoted as $\mathcal{D}$;

- The metric data collected from time $t$ and onward makes up the interventional (anomalous) dataset, denoted as $\mathcal{D}^*$;

This mapping allows us to consider a failure as an intervention on the failing node. Considering the failure as an intervention gives us the ability to leverage the existing literature for finding the interventional target and build a customized solution for RCA, which we explain next.

# 3   Root Cause Analysis

In this section, we formalize the problem and discuss a few challenges that are still open in detecting the root cause of the failure.

**Problem Formulation.** In a cloud system, the problem of finding the root cause of the failure can be formalized as follows. A microservice-based cloud application consists of a set of $n$ microservices, $\mathcal{S} = \{s_1, \ldots, s_n\}$. With a given time interval, the monitoring tool collects at least $m$ metrics from each of the microservices, i.e., $\mathcal{M}(i, t) = \{r_{i,1,t}, \ldots, r_{i,m,t}\}$ where $m \geq 1; \forall i \in \{1, \ldots, n\}$. Here, $\mathcal{M}(i, t)$ is a set of $m$ metrics of microservice $i$ at time instance $t$. To combine this all together, we have two time series datasets defined as $\mathcal{D} = \{\mathcal{M}(1, 1), \ldots, \mathcal{M}(n, t - 1)\}$ and $\mathcal{D}^* = \{\mathcal{M}(1, t), \ldots, \mathcal{M}(n, \mathcal{T})\}$ where $t$ represents the time when the failure was first registered and $\mathcal{T}$ is the time when the bug was fixed. We also assume that all the columns have discrete values as the continuous values can be discretized if needed.

The problem of localizing the failure is difficult because of the high noise-to-signal ratio because of failure propagation. A failed service will affect all the parent services and therefore it becomes difficult to find the culprit. There has been some recent literature that tries to narrow down the root cause using techniques from causal inference [34, 13, 6, 10, 16, 28], however, it has a few shortcomings.

**Domain knowledge and parametric assumptions.** For simplification, most existing works make parametric assumptions that might not hold in the real-world [28, 10, 34]. For instance, consider a case where the latency of a microservice grows linearly to the CPU utilization. However, after a point, the degree of parallelism exceeds the gain of running multiple requests in parallel (because of the increasing cost of context switches), after which the latency increases exponentially with respect

to the number of incoming requests. This non-linearity has been pointed out myriad times in the systems literature [36, 15, 2].

Moreover, some studies use domain knowledge to construct a causal graph [6, 10] that limits their applicability and works only in strict cases. For example, one of the rules used in CIRCA [10] is that the callee's traffic load affects the caller's latency. This statement holds only when the communication between two services is a blocking call. If we have a non-blocking call where the caller does not wait for the response from the callee, this statement might lead to the wrong dependency graph. In contrast, our proposed solution does not make any parametric assumption nor requires any domain knowledge to find the root cause.

**Checking dependence between variables.** One straightforward way to check if a service is the root cause or not is to observe how much metrics of that service were affected after the failure. This is the strategy followed by $\epsilon$-Diagnosis [23]. It uses a modified form of the coefficient of variation (COV) to check if a metric changed significantly during the failure. The critical problem with pairwise distance measures such as COV is that these measures do not condition on other variables which leads to a high false positive rate.

A common way to overcome the limitation of pairwise measures is to use conditional independence (CI) tests. To this extent, most existing works use some version of the PC algorithm [19] to construct the dependency graph between the services [4, 30, 13, 6, 16]. There are two major problems with the PC algorithm. First, the PC algorithm only works with observational data and is therefore unable to use the information presented in the interventional data. This can lead to incomplete causal graphs (with observational data, we might only be able to find the correlation between two variables but with interventional data, we can infer the cause and effect). Second, the PC algorithm tries to learn the complete underlying causal graph which requires a higher number of conditional independence tests to be executed therefore making it impractical for applications with a large number of services.

However, our method is capable of learning the causal structure by systematically leveraging the distributional invariance present not only within the observational data but also across the observational and interventional datasets. This has been proven to be strictly more informative than just using the observational dataset [20, 8]. Moreover, our proposed approach tries to find the root cause without learning the whole causal graph, therefore, reducing the number of CI tests and making PC practical for large-scale cloud applications.

## 4 Hierarchical Learning for RCA

We make the key observation that in a cluster of microservices, a failure can be thought of as performing a soft intervention on the failing node by changing its generative mechanism. Consider the latency of a microservice as a random variable $L$ in a causal graph. During the normal state, it follows the probability distribution $P(L|Pa_L)$, where $Pa_L$ is its parents in the graph. However, because of a programming bug or a configuration change, the probability distribution of the latency changes to $P' \neq P$. Such an effect (failure) can be considered a soft intervention as the affected node remains connected to its parents, but its conditional distribution changes. Accordingly, we define a variable $R$ as a root-cause variable if $P(R|Pa_R)$ varies from normal mode of operation to the anomalous mode of operation. The key benefit of modeling the failure as an intervention is that it enables us to translate the problem of finding the root cause of failure into finding the interventional target. This view allows us to leverage the recent developments in the causal discovery literature that can identify intervention targets.

**Learning Interventional Targets.** We picked $\Psi$-FCI [8] which proposes the state-of-the-art sound and complete algorithm to learn the causal graph as well as the interventional targets from the observational and soft-interventional data. $\Psi$-FCI considers the case where the latent confounders are present in the data. Accordingly it uses a modified version of the FCI algorithm [27] to learn the causal graph. However, in our work we assume no confounder variables and therefore running FCI is not necessary. Accordingly, we designed a modified version of $\Psi$-FCI, named $\Psi$-PC, which internally uses PC [27], to find the interventional target considering no latent confounders .

$\Psi$-FCI, and consequently $\Psi$-PC introduces an extra node called the F-NODE to represent the effect of an intervention on the system. Such F-NODE's have been used for this purpose in Pearl's work [19] as a tool to prove some of the do-calculus rules. They were also used for causal discovery in

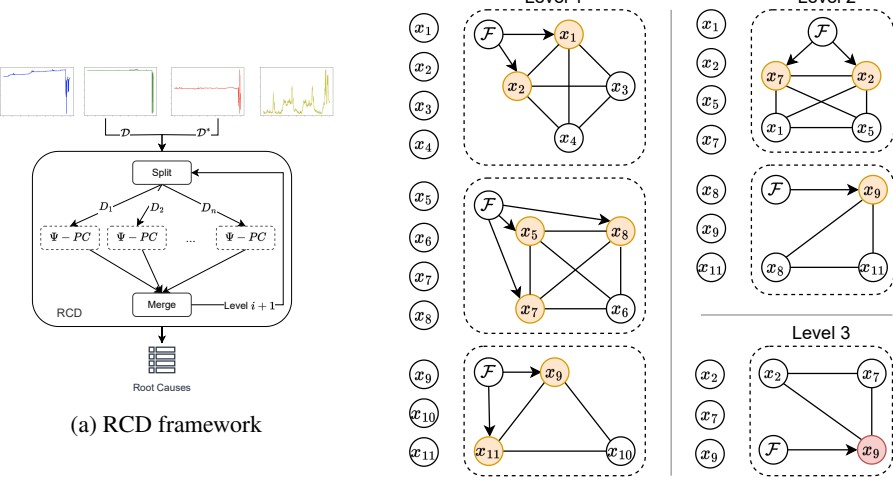

(a) RCD framework

(b) An execution of RCD with 11 nodes and $\gamma = 4$

Figure 1: The hierarchical and localized learning algorithm of RCD. It follows the divide-and-conquer approach by first splitting the dataset into small subsets to find interventional targets from each subset by running $\Psi$-PC (dashed boxes). In the merge phase, RCD combines the candidate root causes of all the subsets and performs the same steps recursively with the new set of variables. It stops when the set of candidate root causes cannot be trimmed further. Figure 1b shows an execution of RCD with 11 nodes where the orange nodes are potential root causes that are carried to the next level for further processing and the red node ($x_9$) is the eventual root cause. Note that RCD only estimates the neighborhood of F-NODE and leaves the rest of the graph untouched therefore reducing the number of CI tests significantly.

several other studies, such as [17, 32]. The utility of such a representation is that one can identify distributional invariances of the form $P_N(X|Pa_X) = P_A(X|Pa_X)$, where $P_N$ and $P_A$ are the distributions under the normal mode of operation and anomalous mode of operation, respectively. This can be done through conditional independence tests involving the F-NODE systematically by leveraging FCI or PC algorithms on the graph involving the F-NODE. Specifically, let us introduce the probability distribution $P^*$ as $P^*(V|F = 0) = P_N(V)$ and $P^*(V|F = 1) = P_A(V)$, where $V$ is the set of observed variables and $F$ is the F-NODE. Then the invariance $P_N(X|Pa_X) = P_A(X|Pa_X)$ corresponds to the conditional independence statement $X \perp\!\!\!\perp F|Pa_X$, which can be tested on $P^*$. Samples from $P^*$ can be obtained by simply concatenating normal and anomalous datasets with an $F$ node that takes the value of 0 for normal and the value of 1 for the anomalous mode of operation. Running the PC algorithm [27] on this combined data with the domain knowledge that F-NODE cannot have any incoming edges allows us to discover all and only the root cause nodes, i.e., $R$ such that $P_N(R|Pa_R) \neq P_A(R|Pa_R)$ as the remaining neighbors of F-NODE. The soundness and completeness of $\Psi$-PC follow from [8]. The detailed algorithm is presented in the appendix.

**Hierarchical Learning.** We propose a hierarchical learning algorithm called RCD that uses $\Psi$-PC as a tool to quickly find the interventional target. Figure 1a provides an overview of the hierarchical learning algorithm. Our key insight is that rather than finding the interventional target using all variables, we can divide the data into small subsets of variables, and find the interventional targets in every subset using $\Psi$-PC. The output of $\Psi$-PC is candidate interventional targets in each subset. With small sets, we reduce the number of CI tests and hence the run time. We use parameter $\gamma$ to control the subset size. Note that partitioning the variables into subsets could lead to the descendants of the root cause and the actual root cause falling into different subsets. After running $\Psi$-PC on a subset, F-NODE will either point to the actual root cause if it exists in that subset or any descendant of the root cause that is not separable by conditioning on the nodes in that subset.

After running $\Psi$-PC on all the subsets, we take the union of the interventional targets from all subsets as potential root causes. Specifically, these are the variables that change their distribution from normal to the anomalous dataset, and whose change cannot be controlled for using the variables in that subset. However, it is possible that had we conditioned on variables in the other subsets, they would be conditionally independent of the F-NODE, and hence be removed from its neighborhood. In other words, the union of interventional targets will have false positive root causes. Accordingly, we can create new subsets using only the nodes in the current neighborhood of F-NODE and repeat this

process. At the final level, we will have a single set of nodes and running $\Psi$-PC on these reveals the actual root causes. Complete pseudo-code of the algorithm is given in Algorithm 1.

---

**Algorithm 1** Root Cause Discovery Algorithm (RCD)

**Input:** Normal dataset $\mathcal{D}$, anomalous dataset $\mathcal{D}^*$, $\gamma$, $\Psi$-PC (.) [8], $k$ : Max. no. of root causes
**Output:** A list of root causes $\mathcal{U}$.

1: **procedure** RCD($\mathcal{D}, \mathcal{D}^*, \gamma$)
2:     $\mathcal{U} \leftarrow$ Set of variables of $\mathcal{D}, \mathcal{D}^*$.
3:     **while** $|\mathcal{U}| > k$ **do**
4:         $\mathcal{S} \leftarrow$ A random partitioning of $|\mathcal{U}|$ into subsets of size $\gamma$.
5:         $R \leftarrow \emptyset$
6:         **for all** $S \in \mathcal{S}$ **do**
7:             $G \leftarrow \Psi\text{-PC}(\mathcal{D}[S], \mathcal{D}^*[S])$         # $\Psi$-PC constructs a graph on $S \cup \{\text{F-NODE}\}$.
8:             $R \leftarrow R \cup Ne_G(\text{F-NODE})$         # Extract neighbors of F-NODE.
9:     $\mathcal{U} \leftarrow R$.
10: **return** $\mathcal{U}$

---

Theorem 1 and its proof in the appendix states that RCD is sound for detecting the root causes.

**Theorem 1** *Given access to a perfect conditional independence oracle, and under the causal sufficiency, and the extended faithfulness[1] assumptions Algorithm 1 returns the true root cause variables.*

**Localized Learning.** In the previous section, we discussed using RCD to find the root cause of the failure using $\Psi$-PC in a hierarchical fashion . Note that $\Psi$-PC focuses on learning not only the interventional targets but also the underlying causal graph. This is useful when some downstream task requires full causal structure [7, 33]. The time complexity of such an algorithm depends on the number of CI tests that need to be executed, which in general is exponential in the number of nodes for non-sparse graphs. By decoupling learning the causal graph from finding the interventional targets, we can improve run time by significantly reducing the number of CI tests.

In our case, to find the root cause of the failure, we only need to learn the interventional target which translates to only learning the immediate neighborhood of the F-NODE. This key insight allows us to propose a localized learning algorithm that focuses on only learning the neighborhood of the F-NODE. The localized learning algorithm only runs the minimal set of CI tests needed to remove non-root cause nodes from the neighborhood of F-NODE. We can accomplish this by modifying $\Psi$-PC such that it only execute the CI test where one of the variables under consideration is F-NODE and therefore the rest of the graph will stay complete . Figure 1b shows an example of RCD with both hierarchical and localized algorithms. The localized version of the algorithm is used in our experiments, presented in the next section.

# 5   Evaluation

We conduct extensive evaluation of our proposed solution to answer the following questions: (1) *How effective is RCD for finding the interventional target?* and (2) *How quickly can RCD find the interventional target?* We report more detailed experiments in the appendix.

**Implementation and Testing Setup.** We implemented a modified version of RCD (Algorithm 1) for finite number of samples that generates a list of top-$k$ potential root causes based on the p-values of CI tests. The complete algorithm with implementation details are provided in the Appendix. We implemented $\Psi$-PC in Python using the causal-learn package[2]. We used the Chi-squared test to check the independence between two variables. To generate synthetic data, we used pyAgrum[3] with a randomly generated DAG to draw samples for the normal and anomalous dataset. For all experiments, we set $\gamma$ to 5 for RCD in all our experiments unless specified otherwise. Finally, to get statistically significant results, we ran all the experiments 100 times and plotted the average. Our source code is available at https://github.com/azamikram/rcd.

---

[1]The distributional invariances across datasets cannot be coincidental, please see [8] for a formal definition.
[2]github.com/cmu-phil/causal-learn
[3]pyagrum.readthedocs.io/en/1.0.0/

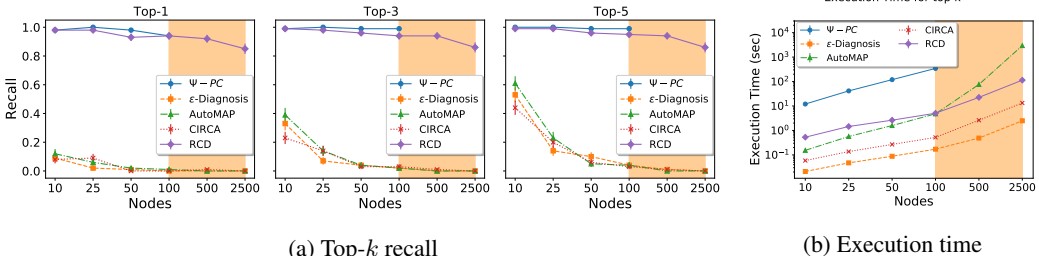

(a) Top-$k$ recall          (b) Execution time

Figure 2: The left figure plots the top-$k$ recall and the right one shows the execution time of different fault localization algorithms. RCD achieves similar top-$k$ recall as $\Psi$-PC and it outperforms the other baselines because they either require domain knowledge, human intervention, or rely only on lower-order tests such as cross covariance. The execution time required for RCD increases slower with the number of nodes than $\Psi$-PC (note the logarithmic scale on the Y-axis), which does not terminate in a reasonable time for 500 and more nodes. AutoMAP performs better than $\Psi$-PC but still incurs a high execution time because it internally uses PC. $\epsilon$-Diagnosis achieves the best execution time but its usefulness is limited as its recall drops significantly when the number of node increases.

**Baseline and Evaluation Metrics.** We compare our solution to the following baseline methods;

- $\epsilon$-Diagnosis [23]: The authors used the coefficient of variation (COV) on the normal and anomalous data to find microservices that changed significantly after the failure. The comparison with $\epsilon$-Diagnosis will show the importance of our use of the CI test.

- AutoMAP [13]: AutoMAP constructs a weighted causal graph using a modified version of PC algorithm where the edge weight between the two microservices signifies the (in)dependence of a performance metric. It concludes by traversing nodes based on a random walk algorithm and finding the root cause from the weighted causal graph using the correlation metric.

- CIRCA [10]: CIRCA uses service call graph to construct a causal graph between the metrics with a set of static and domain-specific rules. It runs a regression hypothesis test on the anomalous data to find the deviation of a variable from its expected normal distribution after the failure. The comparison with CIRCA shows that tools that make parametric assumptions are applicable only under strict constraints and hence fall short in real-world applications.

- $\Psi$-PC: The comparison with $\Psi$-PC will highlight that learning the complete causal graph is not necessary for finding the root cause. Furthermore, it will also establish the time gain RCD gets from localizing learning.

- RCD: The proposed hierarchical solution with localized learning.

Contrary to RCD, CIRCA requires a call graph to build a causal structure between the microservices. For this purpose, we used the output of PC as the call graph for CIRCA. Two quantitative metrics are generally used to evaluate the performance of different algorithms related to root cause detection: execution time and recall at top-$k$ [13, 31, 16, 30, 10]. In our case, recall at top-$k$ is the probability of getting the correct root cause from the top $k$ outputs of the algorithm.

**Finding Root Cause** Our first experiment compares the detectability of different RCA algorithms. We ran this experiment by controlling the number of nodes in the DAG. The max in-degree of all the DAGs are set to 3 and the total number of states for every node is 6. Furthermore, for every experiment, we draw 10K samples for the normal and anomalous states of the system.

Figure 2a shows the top-$k$ recall with $k = 1/3/5$. In general, we see that RCD can identify the interventional target (root cause) with very high recall. More specifically, it achieves 98% recall at top-1. In terms of other algorithms, $\Psi$-PC performs similarly to RCD (but cannot run beyond 100 node). This is because RCD internally uses a modified version of $\Psi$-PC to learn the interventional target, but solves the scalability problem of $\Psi$-PC. On the flip side, recall of AutoMAP for $k = 5$ is just above 50% when there are only 10 nodes and it drops to 0 when the number of nodes increases. One of the key reasons for this result is that the accuracy of AutoMAP depends on the historical data about the failures. Hence, for the initial rounds of the execution (i.e., without expert human input), AutoMAP performs worse as it only relies on the correlation between the variables. $\epsilon$-Diagnosis performs much worse than RCD because of its use of COV which does not take into account other

variables when deciding the independence between two variables. $\epsilon$-Diagnosis performs worse with a larger number of nodes as there is more room to miss the root cause and classify a wrong microservice to be the root cause simply because that metric shows a bigger change. Finally, we observe that CIRCA is unable to find the root cause with higher number of nodes. This is because our data generation model does not follow any specific parametric distribution which makes it difficult to find the root cause using tools that rely on regression-based techniques.

**Execution Time.** Most microservice-based cloud applications are latency sensitive thus any tool that claims to find the root cause of the failure has to minimize the execution time. To measure the execution time of different algorithms, we conducted experiments on synthetic data with a varying number of variables and drew 10K samples for both datasets (normal and anomalous). We measured the time it takes to find the top-$k$ potential root causes where $k = 1/3/5$. Figure 2b illustrates the results.

The first thing to note here is that RCD outperforms all the baselines. On average, for 500 nodes, it can detect the actual root cause of the failure in the top 5 candidates in just above 22 seconds. Secondly, the execution time of $\Psi$-PC grows exponentially as the number of nodes increases. For just 500 nodes, single execution of $\Psi$-PC was taking more than 150 minutes. Here we can observe the benefits of using a hierarchical and localized approach to learn the root cause. Because of the time constraint, we only ran $\Psi$-PC for a maximum of 100 nodes.

Considering other baselines, we see that AutoMAP still incurs a high delay because of its dependence on PC algorithm which tries to learn the whole causal graph. Furthermore, the execution time of $\epsilon$-Diagnosis almost stays constant. The reason for this is that it only considers variables in a pairwise fashion, thus, the number of tests for COV grows only quadratically with the number of nodes. The consequence of only considering the variables in a pairwise fashion is that it leads to a higher number of false positives which significantly affects the top-$k$ recall. Similarly, inference from CRICA is fast because it does not build a causal graph but requires the call graph as an input[4].

The key takeaway from our experiments is that RCD provides almost identical top-$k$ recall as compared to $\Psi$-PC while reducing the execution time significantly. RCD gains this benefit by (1) exploring the data in a hierarchical fashion and (2) by only learning the neighborhood of F-NODE rather than learning the complete underlying causal graph.

# 6 Case Study

**Sock-shop.** To test the applicability and quantify the performance gain of different root cause discovery algorithms, we setup a test-bed using a real microservice-based application. We use Sock-shop [24], a replica of an online application that sells socks. It consists of 13 microservices written in different technologies and each microservice is deployed on its separate VM/container. The communication between the microservices happens through API requests over HTTP. Moreover, all these microservices provide a large amount of statistical information in terms of different metrics (for instance, CPU and memory utilization, latency, and the number of errors). We also developed an independent workload generator using Locust [5] to send the traffic to the Sock-shop application. To mimic a real application, we followed sinusoidal distribution with a mean of 50 to draw the number of users for every second. Every user first signs up on the system, browse a few items, and finally orders one. This whole process sends multiple requests to all the microservices.

There are two very common failures in cloud computing applications: CPU hog and memory leak [4, 14, 13]. We inject these two types of failures in microservices by running `stress-ng` [6]. We accomplish this by modifying the Docker images of all the microservices to install `stress-ng` and then running it on the container of the targeted microservice. With this setup, we injected failures in all of the major microservices of Sock-shop (carts, catalogue, orders, payment, and user). Out of the 13 microservices, we selected these because they were the critical user-facing services. Any change to their performance cause the other microservices to get affected as well, thus making it difficult to detect the root cause with a simple threshold-based scheme.

---

[4]For CIRCA, we did not include the time for running PC to construct the call graph
[5]locust.io/
[6]wiki.ubuntu.com/Kernel/Reference/stress-ng

Table 1: Top-$k$ recall of different algorithms for detecting the root cause of data from the Sock-shop application. $\Psi$-PC suffers mainly because of the limited number of samples whereas RCD can achieve reasonable recall due to localized search. AutoMAP performs worse because of its reliance on the historical data.

| | | CPU hog | | | | Memory leak | | | |
|---|---|---|---|---|---|---|---|---|---|
| | | $\Psi$-PC | AMAP | $\epsilon$-Diag. | RCD | $\Psi$-PC | AMAP | $\epsilon$-Diag. | RCD |
| Top-1 | Carts | 0.6 | 0.05 | 0.2 | **0.69** | 0.0 | 0.14 | 0.4 | **0.6** |
| | Catalogue | 0.0 | **0.22** | 0.2 | 0.17 | 0.0 | 0.0 | **0.2** | 0.11 |
| | Orders | 0.0 | 0.01 | 0.2 | **0.29** | 0.0 | 0.26 | 0.2 | **0.45** |
| | Payment | 0.0 | 0.07 | **0.6** | 0.17 | 0.0 | 0.01 | **0.4** | 0.17 |
| | User | 0.0 | 0.29 | 0.2 | **0.46** | 0.2 | 0.08 | 0.2 | **0.71** |
| Top-3 | Carts | 0.6 | 0.23 | 0.4 | **0.86** | 0.0 | 0.2 | 0.4 | **0.88** |
| | Catalogue | 0.0 | **0.22** | 0.2 | **0.22** | 0.0 | 0.0 | 0.2 | **0.33** |
| | Orders | 0.0 | 0.36 | 0.2 | **0.66** | 0.0 | 0.37 | 0.2 | **0.56** |
| | Payment | 0.0 | 0.07 | **0.6** | 0.4 | 0.0 | 0.05 | **0.8** | 0.34 |
| | User | 0.0 | 0.53 | 0.2 | **0.66** | 0.2 | 0.27 | 0.2 | **0.84** |
| Top-5 | Carts | 0.6 | 0.24 | 0.4 | **0.86** | 0.0 | 0.25 | 0.4 | **0.88** |
| | Catalogue | 0.0 | **0.23** | 0.2 | 0.22 | 0.0 | 0.0 | 0.2 | **0.33** |
| | Orders | 0.0 | 0.51 | 0.2 | **0.66** | 0.0 | 0.49 | 0.2 | **0.56** |
| | Payment | 0.0 | 0.07 | **0.6** | 0.40 | 0.0 | 0.11 | **0.8** | 0.34 |
| | User | 0.0 | 0.55 | 0.2 | **0.66** | 0.2 | 0.3 | 0.2 | **0.84** |

We ran the Sock-shop application for 5 minutes to collect the data for the normal state of the system. After that, we injected the failure and executed the application for 5 more minutes to collect the anomalous dataset. We repeated the experiment 5 times for two different types of failures (CPU hog and memory leak) and in total, we gathered 50 datasets. To measure the performance of the root cause detection algorithm, we ran the algorithm 100 times on every dataset and quantified its top-$k$ recall. In this context, the top-$k$ recall can be interpreted as the probability that the algorithm can find the actual root cause in a list of top-$k$ potential root causes.

Table 1 shows the top-$k$ recall of different algorithms for detecting the root cause of the data collected from Sock-shop. We report the execution time in Table 1 of appendix. RCD performs better than the baselines in general. Considering baselines, $\Psi$-PC performs poorly for most of the microservices. The main reason for this is the limited number of samples for the normal dataset, which makes $\Psi$-PC unable to learn the required causal structure. In a few cases, we observe that $\epsilon$-Diagnosis performs better than RCD because of the fewer number of services available in Sock-shop. However as shown earlier from our experiments with synthetic data, $\epsilon$-Diagnosis does not scale well with large number of nodes.

**Real Data.** We further validate RCD on a set of real-world datasets collected from a part of a production-based microservice system hosted on AWS cloud-native system. The data was collected from Grafana, a monitoring tool for cloud applications. The system consists of 25 different microservices. A total of 150 metrics were collected which broadly cover all the monitoring interfaces of the services. The type of metrics ranged from throughput, memory utilization, service latency to I/O read-writes and system load, *i.e.*, covering the golden signals [3].

We collected data for three distinct outages that occurred in the last 6 months in the system. For every failure, a team of Site Reliability Engineers (SREs) report the time when the outage occurred and when it was fixed. Accordingly, we collected anomalous data for the duration of the failure whereas the normal data contains the metric information for 2 days before the failure. The summary of the outages and the result from RCD and $\epsilon$-Diagnosis are shown in Table 2. $\Psi$-PC was unable to complete its execution within an hour and hence we exclude it. AutoMAP requires a front-end service and needs the same set of metrics to be available for all the services which were not true in our case. Finally, CIRCA requires the metrics to be classified into the metric categories and a call graph which were unavailable to us and hence we exclude it as well. On the flip side, RCD does not require any such information or domain knowledge. Next, we discuss the results of every failure in detail.

**Outage $\mathcal{A}$.** According to the incident report, the outage lasted for over an hour. It occurred during a planned database maintenance that involved shunning and rebuilding database of every node individually from a cluster. However, during the rebuilding phase, AWS autoscaling was not able to provision a similar instance in the same zone due to capacity issues. This resulted in an imbalance

Table 2: The summary and top-7 recall of $\epsilon$-Diagnosis and RCD on data collected for three outages from a production-based cloud application. The length of the vector represents the number of services that were flagged as root cause and the individual number shows the number of metrics belonging to a particular service. The green color illustrates that algorithm was able to correctly detect the root cause whereas the red color shows the algorithm could not find the root cause.

| | | | Rank of Services from top-7 | | Time (sec) | |
|---|---|---|---|---|---|---|
| Outage | Metrics | Duration (min) | $\epsilon$-Diagnosis | RCD | $\epsilon$-Diagnosis | RCD |
| $\mathcal{A}$ | 137 | 65 | [**1**,1,1,1,1,1,1] | [**3**,1,1,1,1] | 0.145 | 112 |
| $\mathcal{B}$ | 147 | 72 | [**2**,1,1,1,1,1] | [**3**,1,1,1,1] | 0.186 | 239.8 |
| $\mathcal{C}$ | 150 | 210 | [**3**,1,1,1,1] | [**4**,1,1,1] | 0.146 | 22.57 |

in autoscaling which trickled down as a series of faults. On running RCD on this dataset, it found the memory footprint of different tiers of the database as the top-2 root causes, while the resource utilization of the database for the message queuing system was flagged within the top-7 root causes.

**Outage $\mathcal{B}$.** During this outage, which lasted for about an hour and 12 minutes, the service instances located at a specific region were unavailable affecting several customers. The system failed due to a fault in the event consumer queue which got stuck in one of the microservice. The consequence was longer database query time and increased memory size in the application server. Executing RCD with the outage data found the heap size and the system load of the event queue to be the culprit among the top-3 root cause. However, the top-1 root cause, which was the hit ratio in the Memcached database was not among the faulty services. We believe, there might have been some latent variable between the failed service and the Memcached database that was affecting both services. We leave the problem of finding root causes with latent variables for future work.

**Outage $\mathcal{C}$.** In another outage that lasted for around 3.5 hours, alerts regarding high error rates in a microservice (donated as $\mathcal{M}$ here for simplicity) were fired. The users were facing errors while accessing $\mathcal{M}$ however, the team of SREs found no issues with the services. Later it was found that a faulty update in one of the AWS components has caused that component to fail. That component was being used by $\mathcal{M}$ and therefore after the update $\mathcal{M}$ was unable to process incoming requests. Note that in this outage, the system was not collecting metrics information about faulty AWS components because it was out of the boundary of the system. However, running RCD on the metrics data available, it found the disk space, error rate, and heap size of $\mathcal{M}$ to be the root cause metrics. Even though the faulty component was outside the system, RCD was able to find the directly affected service as the root cause. This type of information is immensely useful for SREs to narrow down the problem during system diagnosis.

## 7 Conclusion

We proposed a novel root cause analysis algorithm that systematically leverages the distributional invariances across normal and anomalous datasets, taking inspiration from the recent causal discovery algorithms. Our tailored algorithm sidesteps learning the full causal graph and rather focuses on only the root causes. This provides not only significant benefits in runtime, but also in the number of required anomalous samples.

## 8 Acknowledgement

This material is based in part upon work supported by Adobe Research, the Army Research Office under Contract number W911NF-2020-221, and the National Science Foundation under Grant Numbers CNS-2016704 and CCF-1919197. Any opinions, findings, and conclusions or recommendations expressed in this material are those of the authors and do not necessarily reflect the views of the sponsors. We further appreciate the anonymous reviewers for their valuable and constructive feedback that greatly improved the manuscript. We would also like to thank Ahaan Dabholkar for assistance with running and automating the experiments and Shubham Agarwal for assistance with querying real enterprise data.

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
