# Root Cause Analysis of Failures in Microservices through Causal Discovery

**Azam Ikram**[1]    **Sarthak Chakraborty**[2]    **Subrata Mitra**[2]    **Shiv Kumar Saini**[2]
**Saurabh Bagchi**[1]    **Murat Kocaoglu**[1]
[1]Purdue University, USA    [2]Adobe Research, India
{mikram,sbagchi,mkocaoglu}@purdue.edu
{sarchakr,sumitra,shsaini}@adobe.com

## Appendix

This section of the paper discusses some details about root cause analysis and RCD in details.

**Preliminaries**

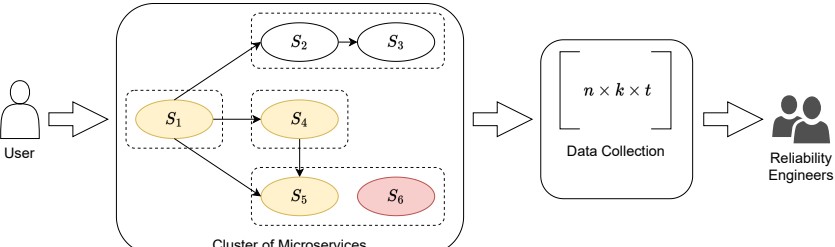

Figure 1: The workflow of a microservice-based system from the usage point of a client and a Site Reliability Engineer (SRE). Every oval is a microservice whereas every dashed rectangle is a host/VM that hosts multiple microservices. The edges between two microservices represent the call graph. Red-colored microservices signify a failure but the effect is observed on the yellow-colored microservices. Even with the topological information about the microservices, it will be difficult for a SRE to pinpoint the root cause just observing the failures at, say $s_5$.

**Microservice Architecture.** Microservices are usually containerized applications that perform a small but specific set of tasks independently from other microservices. In a microservice-based application, a set of microservices work together to fulfill a client's request. For monitoring and debugging purposes, all of the microservices expose some kind of performance metrics. There are many tools that help developers instrument the microservices to expose useful information as well as collect, store and visualize all that data. Most of these tools scrap all the microservices periodically, store the metric information, and produce a time-series dataset. SREs use this time-series metric data to monitor the performance of the overall application as well as the health of individual microservices. Figure 1 depicts a general web application with multiple microservices running on different hosts.

**Causality and Causal Graph.** We define causality as a means to measure the dependence between two variables. One way to depict the causal relation between these variables is the causal graph. In the causal graph, every node represents a random variable whereas an edge between the two nodes shows the dependence between the variables. The conditional probability distribution of a node in the causal graph can be computed as

$$P(X_i|pa_i) \tag{1}$$

Here $X_i$ represents the random variable $i$ in the causal graph and $pa_i$ is the set of parents. We can use 1 to compute the marginal probability distribution of a node by summing over the probability of

all the values of $pa_i$.

$$P(X_i) = \sum_x P(X_i|x)P(x) \qquad (2)$$

Moreover, the probability of the overall system with $n$ random variables can be computed as

$$P(X_1, \ldots, X_n) = \prod_i P(X_i|pa_i) \qquad (3)$$

We can use 1 as the conditional independence (CI) test to check the (in)dependence relationship between two variables. For instance, two variables $X$ and $Y$ are independent only if we can find a set $S \cap \{X, Y\} = \emptyset$ such that $P(X|Y, S) = P(X|S)$.

Formally, a causal graph can be defined as a DAG $G(V, E)$ where $V$ is the set of $n$ vertices, $V = \{v_1, v_2, ...v_n\}$, and $E \subseteq V \times V$ is the set of the directed and undirected edges. A directed edge $\vec{e} = \{v_1, v_2\}$ signifies that $v_1$ causes $v_2$. Here the edge $\vec{e}$ shows that we could not find a *separating set*, $S$, such that $P(v_1|v_2, S) = P(v_1|S)$.

**Causal Discovery** Causal discovery is a well-known problem in causal inference and has been studied thoroughly [26, 37, 3, 1]. It has applications in social sciences, bioinformatics, economics, and has recently found its way into machine learning as well [25, 2, 24, 27, 38, 13]. Broadly speaking there are two type of algorithms for causal discovery, **Score-based causal discovery** [6, 29] and **Constraint-based causal discovery** [30, 11]. Score-based algorithms aim to find the causal structure by optimizing a properly defined score function. Among them, Greedy Equivalence Search (GES) [6] is a well-known two-phase procedure that directly searches over the space of equivalence classes.

The constraint-based causal discovery algorithms, on the other hand, start with a complete graph and remove the edges between a pair of variables if they are conditionally independent. To test the independence of two variables, denoted as $X \perp\!\!\!\perp Y$, it uses some well-established statistical conditional independence (CI) tests such as Fisher z-transformation or G-test for hypothesis testing. In contrast to score-based causal discovery algorithms, there have been several studies done for constraint-based algorithms in the presence of interventional data [16, 23, 39].

A renowned constraint-based causal discovery algorithm is the PC algorithm [30]. PC algorithm works in two phases. The first phase is where it tries to learn the skeleton graph (undirected graph) and in the second phase, it tries to orient the edges between the nodes. Starting with a completely connected graph, it picks a connected pair of variables; let us call them $X$ and $Y$, and runs a CI test to check if the variables are dependent or not. If they are independent, $X \perp\!\!\!\perp Y$, it removes the edge between the two variables. Next, if two variables are still connected in the graph, it tries to find a separating set $S$, such that conditioning on that set makes the variables independent. To try to find the smallest $S$, after every iteration, it increases the cardinality of the separating set by 1.

The output of the first phase of the PC algorithm is an undirected skeleton graph, where an edge represents a dependence between the two connecting variables. The next step is to orient the edges of the skeleton graph. The PC algorithm achieves this with a set of rules [20].

From the causal inference literature, there are a plethora of studies that try to learn the underlying causal structure in the presence of observation and interventional data [31, 1, 3, 16, 37]. Furthermore, there are also a set of studies that try to identify the interventional target (the node where intervention happened) [23, 7, 39, 14].

**Proof of Theorem 1.**

Note that $\Psi$-PC concatenates two datasets with an F-NODE and runs PC algorithm on all the variables by treating F-NODE as another variable. Even though with more than two datasets this would create faithfulness violations among multiple F-NODE's as pointed out in [14, 23], no such issue arises when there are two datasets creating only a single F-NODE. Accordingly, $\Psi$-PC can avoid the careful treatment between F-NODE's that $\Psi$-FCI requires. Same for RCD which uses $\Psi$-PC.

We will use $p_N$ to represent the probability distribution of variables under normal operating conditions (observational), and $p_A$ to represent the one under anomaly (interventional). Let us introduce the probability distribution $p^*$ that is defined as $p^*(V|F = 0) = p_N(V)$ and $p^*(V|F = 1) = p_A(V)$, where $V$ is the set of observed variables and $F$ is the F-NODE representing the effect of intervention.

First, note that if $\Psi$-PC was run on the full graph, it would output a graph where the F-NODE is adjacent to only the true root causes. This is because any variable $S$ that is not a root cause can be separated from the F-NODE as $S \perp\!\!\!\perp F \,|\, Pa_S$, since $p_N(s|pa_s, F = 0) = p_A(s|pa_s, F = 1)$ and $Pa_S$ is observable for all $S$ due to the causal sufficiency assumption.

RCD algorithm runs $\Psi$-PC on a subset of variables in every level. F-NODE will then remain adjacent to those that are not separable from it by conditioning on any subset of the variables in that subset. Due to the extended faithfulness assumption, we have that $p_N(s|u, F = 0) \neq p_A(s|u, F = 1)$ for any root cause node $S$, and for any subset $U$ of the observed variables, or equivalently $S \not\perp\!\!\!\perp F \,|\, U$. Therefore, root-causes cannot be separated using any subset of the observed variables. This establishes that, whichever subset the true root causes fall into, they will all remain adjacent to the F-NODE of that subset. This is true for all levels of the algorithm. This establishes the soundness of the algorithm, that at any level, including the final level, root-causes will remain adjacent to the F-NODE.

RCD however is not complete. We have the following simple example to demonstrate that the returned set might contain additional nodes that are not root causes. Consider the augmented graph $F \to R \to S, T \to R, T \to S$. Suppose we partition the nodes into subsets $\{\{T\}, \{R, S\}\}$. F-NODE will not be adjacent to $T$ since $F \perp\!\!\!\perp T$. For the second subset, F-NODE will be adjacent to both $R, S$. This is because one needs to condition on both $R$ and $T$ in order to d-separate F-NODE from $S$. However, $T$ was removed in the first subset and is not included in the next stage. Hence at the end of the execution, RCD will be connected to both $R$ and $S$ even though the actual root cause is $R$.

Please note that one can easily resolve this issue by running one more stage at the end of RCD algorithm by including all the nodes $\{U : F \perp\!\!\!\perp U\}$. These are the nodes that are non-descendants of the F-NODE, which contains the parents of the root causes. Including them enables finding a valid separating set for any node that is not a root cause. This is because $R, Pa_R$ blocks all the backdoor and frontdoor paths from the root causes that would d-connect F-NODE and any non root cause node.

In our experiments we only implemented the vanilla RCD which is sound but not complete as it was sufficient to achieve the recall of $\Psi$-PC. Furthermore, existence of false positives is not a problem for root cause analysis application as long as most of the variables are eliminated quickly as potential root causes in an automated manner.

**Hierarchical Learning Algorithm**

In this section, we provide a modified version of RCD when the number of samples are finite. Similar to Algorithm 1 in the main paper, the algorithm for finite number of samples is divided in two phases as well. In the first phase, we divide the data in small subsets of size $\gamma$ and run $\Psi$-PC to learn the potential interventional targets. In the second phase, we take the union of all the potential interventional targets and narrow the list down to $k$ interventional targets. Complete pseudocode of this algorithm is provided in Algorithm 2.

In most of the RCA literature, the output of the algorithm is an ordered list of potential root causes of size $k$, rather than a singleton. Here $k << n$ and $n$ is the total number of microservices. We create an ordered list of potential root causes by running $\Psi$-PC multiple times with different $\alpha$.

The $\Psi$-PC algorithm takes a hyperparameter $\alpha$ as an input which it uses this to compare the p-values of CI tests to decide if two variables are independent. If the p-value is greater than $\alpha$ then we mark those two variables as independent. In this way, a strict (small) value of $\alpha$ will result in a sparse graph whereas a relaxed (large) value of $\alpha$ will produce a dense causal graph. We create an ordered list of $k$ potential root causes by running $\Psi$-PC multiple times with different values of $\alpha$ and ordering the neighbors of F-NODE.

To create an ordered list of root causes, we modified the second phase of the hierarchical learning algorithm. After running $\Psi$-PC on all the subsets, F-NODE may have more than one neighbor. In this case, we need to decide which one is more likely to be the root cause. We achieve this by choosing the neighbor that is least likely to be the root cause, i.e., a node that would not have been the neighbor of F-NODE only if we had chosen a strict value for alpha. Once we identify that node, we place it at the end of the list, remove it from the neighbors of F-NODE and repeat the process until F-NODE is isolated.

On the flip side, there might be some cases where $k < |Ne_G(\text{F-NODE})|$. In those cases, we will need to find more potential root causes to build a list of $k$ root causes. For this, we start RCD with

**Algorithm 1** Ψ-PC: Algorithm for learning the causal graph and interventional targets from observational and interventional probability distributions [14]

    **Input:** Observational distribution $P_N$, interventional distribution $P_A$, and set of variables $V$
    **Output:** A causal graph on $\mathcal{G}$.
1: **procedure** Ψ-PC($P_N, P_A, V$)
2:     $P^*(V|\text{F-NODE}=0) \leftarrow P_N(V), P^*(V|\text{F-NODE}=1) \leftarrow P_A(V)$         # Concatenate
3:     $P^*(\text{F-NODE}=0) \leftarrow 0.5, P^*(\text{F-NODE}=1) \leftarrow 0.5$       # Equal number of samples
4:     **Phase-1: Learn Skeleton**
5:     $\mathcal{G} \leftarrow$ a complete undirected graph over $V \cup \{\text{F-NODE}\}$
6:     **for all** every pair $X, Y \in V \cup \{\text{F-NODE}\}$ **do**
7:         **for all** $S \subseteq V \setminus \{X\}$ **do**
8:             **if** $P^*(X|Y, S) = P^*(X|S)$ **then**
9:                 $SepSet(X, Y) \leftarrow S$ and remove the edge between $X$ and $Y$ from $\mathcal{G}$
10:     **Phase-2: Edge Orientation**
11:     $\mathcal{R}_0$: For any triplets $< X, Z, Y >$, such that $X \cap Y = \emptyset$, orient $X \rightarrow Z \leftarrow Y$ iff $Z \notin SepSet(X, Y)$
12:     $\mathcal{R}^*$: Orient all adjacent edges of F-NODE as outgoing.
13:     Apply the four Meek rules from [21] ($\mathcal{R}_1 - \mathcal{R}_4$) until none applies.
14: **return** $\mathcal{G}$

---

**Algorithm 2** RCD for finite number of samples

    **Input:** Normal dataset $\mathcal{D}$, anomalous dataset $\mathcal{D}^*$, $\gamma, \alpha, k$ : Max. no. of root causes
    **Output:** A list of root causes $\mathcal{U}$.
1: **procedure** RCD($\mathcal{D}, \mathcal{D}^*, \gamma, \alpha, k$)
2:     $\mathcal{U} \leftarrow$ Set of variables of $\mathcal{D}, \mathcal{D}^*$.
3:     **while** $|\mathcal{U}| > k$ & $|\mathcal{U}| > \gamma$ **do**
4:         $\mathcal{S} \leftarrow$ A random partitioning of $|\mathcal{U}|$ into subsets of size $\gamma$.
5:         $R \leftarrow \emptyset$
6:         **for all** $S \in \mathcal{S}$ **do**
7:             $G \leftarrow$ Ψ-PC($\mathcal{D}[S], \mathcal{D}^*[S], \alpha$)      # Ψ-PC constructs a graph on $S \cup \{\text{F-NODE}\}$.
8:             $R \leftarrow R \cup Ne_G(\text{F-NODE})$         # Extract neighbors of F-NODE.
9:     $\mathcal{U} \leftarrow R$.
10: **return** TOPK($\mathcal{D}[\mathcal{U}], \mathcal{D}^*[\mathcal{U}], k$)       # Get top-$k$ neighbors of F-NODE.

---

a really strict value of alpha (mostly 0.001). If we get at least $k$ neighbors of F-NODE, we stop the execution, order the neighbors based on their p-values, and choose the top $k$ nodes from the list to be the top $k$ root cause candidates. However, if the neighbors of F-NODE are less than $k$ then we rerun the hierarchical learning algorithm with a strict value of alpha and repeat the whole process with a new value of alpha until we get $k$ potential root causes. The top-$k$ algorithm is shown in Algorithm 3.

---

**Algorithm 3** Top-$k$ Root Causes

    **Input:** Normal dataset $\mathcal{D}$, anomalous dataset $\mathcal{D}^*$, Ψ-PC (.) [14], $k$ : Max. no. of root causes
    **Output:** An ordered list of top-$k$ neighbors of F-NODE.
1: $\alpha_0 \leftarrow 0.001, \delta \leftarrow 0.1, \tau \leftarrow 1$
2: **procedure** TOPK($\mathcal{D}, \mathcal{D}^*, k$)
3:     $\mathcal{L} \leftarrow \emptyset, \alpha \leftarrow \alpha_0$
4:     **while** $\alpha < \tau$ **do**
5:         $G \leftarrow$ Ψ-PC($\mathcal{D}, \mathcal{D}^*, \alpha$)
6:         $\mathcal{N} \leftarrow \{x | x \in Ne_G(\text{F-NODE}), x \notin \mathcal{L}\}$     # List of new neighbors of F-NODE.
7:         **for all** $N \in \mathcal{N}$ **do**
8:             $\mathcal{L} \leftarrow \mathcal{L} \cup N[argmax(max(Pv_G(N)))]$       # Sort the neighbors
9:     $\alpha \leftarrow \alpha + \delta$
10: **return** $\mathcal{L}[0 : k]$

---

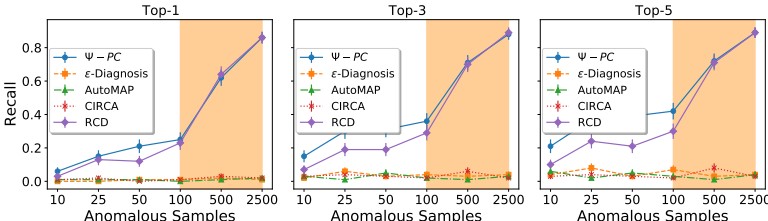

Figure 2: Sample Efficiency of different root cause localization algorithms. As the number of samples grow, the top-$k$ recall of RCD and $\Psi$-PC starts to increase as well. That is to say, RCD can provide the same benefits as $\Psi$-PC while reducing the execution time significantly.

**Implementation and Test-bed Setup**

We implemented Algorithm 2 and $\Psi$-PC in Python using the causal-learn package. For the localized learning algorithm, we re-implemented the first phase of $\Psi$-PC to generate the skeleton graph. Furthermore, we also added the feature to store the p-values of CI tests that we use to rank the root causes. Our source code is available at https://github.com/azamikram/rcd.

We use the Chi-squared test to check the independence between the two variables. To generate synthetic data, we use pyAgrum with a randomly generated DAG by controlling the number of nodes and the maximum in-degree. Next, we randomly populate the conditional probability tables for all the nodes. Using these conditional probability tables, we draw a specific number of samples for the normal dataset. To simulate an intervention on a node, we first select a node at random and randomly modify its conditional probability table thus modifying its probability distribution.

For Sock-shop, we created a cluster of microservices on Amazon Web Services (AWS) using Kubernetes[1]. The cluster consists of 8 t2.xlarge machines (4 vCPU, 16 GB memory) running Ubuntu 18.04. One of these machines is the master node, whereas the remaining are worker nodes. The master node deploys the Sock-shop application on the worker nodes and routes the incoming traffic to the correct microservice. Other than these virtual machines, we have one more t2.xlarge instance running locust to send the traffic to the Sock-shop application.

On top of the Sock-shop application, we have another important tool, Prometheus. We configured it to collect the metric data from the microservices once every second. We collect the workload, CPU usage, memory usage, error counts, and latency for every microservice. In total, we collect 38 metrics from Sock-shop application (not all the microservices expose all metrics such as error counts). Finally, we use this data to generate a time series dataset of the system during the normal and anomalous states.

We injected two types of failures, CPU hog and memory leak, in different microservices of Sock-shop by running `stress-ng`. We accomplish this by modifying the Docker images of all the microservices to install `stress-ng` and then running it on the container of the targeted microservice. With this setup, we injected failures in all of the major microservices of Sock-shop (carts, catalogue, orders, payment, and user).

Sock-shop exposes a lot of metrics but not all of them are equally useful. Some of the metrics, such as the memory utilization of an inactive microservice, might remain constant during the experiment. A general solution for this problem is to remove constant variables from the data in the pre-processing step. We accomplish this by removing a metric that has a standard deviation below 1. Moreover, all of the metrics produce continuous values but the Chi-squared test only works on categorical data. Hence, we discretize the variables into a specific number of *bins* by using K-means clustering. For all our experiments with Sock-shop, we used 5 number of bins.

**Sample Efficiency**

Timely fix of failures in a large scale cloud-based application is of utmost importance. In other words, the root cause of the failure needs to be detected as early as possible. Consequently, one would

---

[1] https://kubernetes.io/

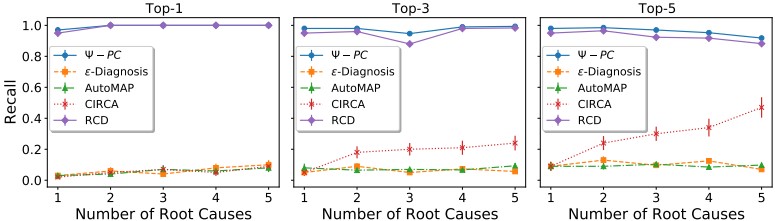

Figure 3: The top-$k$ recall of fault localization algorithms to detect multiple root causes. RCD is as effective as $\Psi$-PC in detecting multiple root causes whereas other baselines fall short. CIRCA performs better than other baselines as it was able to find 47% of the root causes for some cases in the top-5.

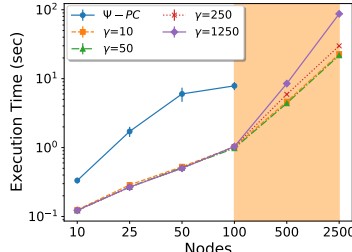

Figure 4: The execution time of $\Psi$-PC and RCD with different $\gamma$. It controls the size of the subsets for RCD, thus increasing $\gamma$ leads to a higher execution time. However, even with higher $\gamma$ and large number of nodes, RCD performs better than $\Psi$-PC because of its localized learning.

expect a large amount of data for the normal state of the system however, the number of samples for the anomalous dataset will be quite limited. To check the applicability of any algorithm for RCA, we need to see how sample efficient that approach is. To that end, we performed an experiment by varying the number of samples and checking the top-$k$ recall of different algorithms. For this experiment, we kept the number of nodes and number of samples for normal dataset constant (100, and 10K respectively) and varied the number of anomalous samples. Figure 2 illustrates the results.

The first thing to note here is that, in most cases, RCD is as effective in detecting the root cause as $\Psi$-PC even with limited number of samples. More specifically, with only 500 anomalous samples, RCD can find the root cause in top 5 with the recall of 71% (as compared to 77% for $\Psi$-PC). Whereas AutoMAP and $\epsilon$-Diagnosis can only detect about 4% and 1.1% of the root causes respectively.

**Multiple Root Causes**

$\Psi$-PC, similar to $\Psi$-FCI, is not limited to only finding a single interventional target as the proposed approach in [14] can find multiple interventional targets from a single dataset. Accordingly, we performed an experiment to analyze the ability of RCD to detect multiple root causes from an anomalous dataset. We kept the total number of nodes to 50 and randomly chose a subset of nodes to be the root cause. Figure 3 illustrates the top-$k$ recall of different approaches. We observe that with higher number of root causes, RCD and $\Psi$-PC are very likely to find a root cause in top-1. However, for top-5 recall RCD's performance start to decrease with 5 root causes (from 1.0 to 0.88). We believe this is because of insufficient number of anomalous samples (the number of anomalous samples were constant for this experiment, 10K). Even still RCD outperforms the best baseline (CIRCA) by 41%.

**Sensitivity analysis of $\gamma$**

RCD requires a hyperparameter, $\gamma$, which controls the size of each subset. In all of our experiments, we set $\gamma$ to be 5. Here, we experiment by changing the value of $\gamma$ to understand how it affects the performance of RCD. We run this experiment by controlling the number of nodes and the value of $\gamma$. The maximum in-degree of all the DAGs is 3, whereas the total number of states for every node is 6. For every experiment, we draw 10,000 samples for the normal and anomalous states of the system.

Table 1: Execution time in seconds of different algorithms for detecting the root cause on data from the Sock-shop application. $\epsilon$-Diagnosis takes less time because of its reliance on coefficient of variation but it performs poorly when the number of variables increase as shown in Figure 2 in the main paper. RCD outperforms $\Psi$-PC and AutoMAP by 97% and 99% respectively.

| | CPU hog | | | | Memory leak | | | |
|---|---|---|---|---|---|---|---|---|
| | $\Psi$-PC | AMAP | $\epsilon$-Diag. | RCD | $\Psi$-PC | AMAP | $\epsilon$-Diag. | RCD |
| Carts | 8.96 | 26.92 | 0.01 | 0.04 | 17.05 | 24.93 | 0.01 | 0.04 |
| Catalogue | 9.41 | 9.64 | 0.01 | 0.05 | 10.51 | 27.45 | 0.01 | 0.04 |
| Orders | 2.96 | 15.44 | 0.01 | 0.04 | 1.97 | 15.20 | 0.01 | 0.04 |
| Payment | 12.40 | 21.48 | 0.01 | 0.04 | 30.86 | 25.17 | 0.01 | 0.05 |
| User | 1.34 | 15.46 | 0.01 | 0.04 | 3.76 | 12.10 | 0.01 | 0.05 |

We run every experiment 100 times and plot the average completion time of RCD. We also plot the execution time of $\Psi$-PC as the reference point. Figure 4 shows the result.

The first thing to note is that with a small number of nodes, different values of $\gamma$ produce the result at about the same time. That is because when $n < \gamma$ where $n$ is the number of nodes, the value of $\gamma$ does not affect the number of subsets. In this case, there will only be one subset for the whole dataset. The effects of $\gamma$ become more visible with a larger number of nodes and for the higher values of $k$. With the larger number of nodes, RCD creates more subsets if $\gamma$ is small thus reducing the overall time. Even with higher values of $\gamma$, RCD performs better than $\Psi$-PC owing to its localized learning scheme. The time for higher values of k increases because we have to run multiple executions of Algorithm 3 to get a sorted list of all the neighbors of F-NODE.

**Sock-shop Experiment**

We ran the Sock-shop application for 5 minutes to collect the data for the normal state of the system. After that, we injected the failure and ran the application for 5 more minutes to collect the anomalous dataset. We repeated the experiment 5 times for two different types of failures (CPU hog and memory leak) and in total, we gathered 50 datasets. To measure the performance of the root cause detection algorithm, we ran an algorithm 50 times on every dataset and quantified their top-$k$ recall and execution time. Table 1 in the main paper shows the top-$k$ recall and Table 1 shows the execution time of different RCA algorithms.

The first thing to note here is that $\epsilon$-Diagnosis outperforms all the other algorithms. That is because it only uses a pairwise test of coefficient of variation to find the root cause metric and microservice. Another reason it takes much less time is that for Sock-shop, we only have 38 metrics. However, as we show in our experiments in the main paper, the recall of $\epsilon$-Diagnosis drops significantly because of the weak CI test for a large number of variable.

Considering RCD, we observe that it outperforms $\Psi$-PC and AutoMAP. More specifically, it reduces the time to find the root cause by 91% and 96% as compared to $\Psi$-PC and AutoMAP in the case of CPU hog failures. We get this gain by using a hierarchical approach to learn the root cause and only learning the neighborhood of F-NODE rather than learning the complete causal graph.

**Related Works**

Performance diagnosis in a distributed system is of paramount importance due to the higher failure chances of each component, hence affecting the system in its entirety. There has been significant research work not only in academia [28, 5, 34] but in the industry [32, 33] as well devoted to the topics of identifying and localizing root causes for the system failures. Works on root cause analysis can broadly be segregated into the following methods:

**Log-based.** These works make use of system log information to investigate the causes of anomalies and failure of a system [9, 36, 40]. A general approach to log analysis involves parsing the raw text data to learn historical patterns, and leveraging the patterns learned to detect anomalies. [9] uses finite state automata to represent the patterns, whereas a recent study [40] performs a learning-based methodology on the features extracted from logs to detect anomalies. The log-based approaches assume that the application produces useful logs. However, in reality the quality and quantity of

different modules of the same applications can be poles apart thus, making them useful only in limited space.

**Trace-based.** Studies like Pinpoint [4], X-Trace [8], and Microscope [18] record the execution path information and locate the culprit services. While Pinpoint and X-Trace require instrumenting the source code and a general understanding of the code by system administrators, Microscope aims at generating a service causal graph based on the trace data. Seer [10], on the other hand, uses RPC-level tracing along with low-level hardware monitoring to identify the root cause microservice, while [17, 19] leverages unsupervised trace anomaly detection and microservice localization to detect the root cause. Furthermore, GMTA [12] uses a graph-based approach to cluster the traces. It also builds an interpretable tool for understanding microservice architecture and diagnosing problems (it does not explicitly work on identifying the root causes). One common trait of most of these tools is that they require application instrumentation and expert knowledge therefore making them difficult to use in real-world applications.

**Metrics-based.** These tools aim at performing system diagnosis by leveraging the value of various metrics observed. MonitorRank [15] localizes the root cause API of failures by examining the historical time series of performance metrics along with the call graph of services. Grano [32] deployed at Ebay Inc. uses a dependency graph-based approach among physical resources to perform root cause analysis. With a call graph, we can analyze the faulty and the impacted services. However, to understand why a fault occurred, we need to understand how performance metrics for each component interact with each other. Works like [34, 35, 5, 22, 28] perform root cause analysis by building causal graphs using variations of the PC algorithm at the performance metrics level. The causal discovery algorithms output a CPDAG where some edges are undirected, and hence post-processing is necessary to convert the CPDAG to a DAG. After generating a DAG, the most common approach is to run a variety of random work or graph traversal algorithms for ranking a subset of probable root causes. These tools do not require any application instrumentation or expert knowledge. However, they suffer from high execution times because of PC that tries to learn the complete underlying causal graph.

On the contrary, our proposed algorithm models the failures as an intervention on the failing node. Thus, allowing us to leverage the state-of-the-art algorithm from causal inference to learn the interventional target. Furthermore, we propose localized hierarchical learning to overcome the high execution time of PC, therefore, making it applicable in real-world applications.