# OpenReview forum: "Root Cause Analysis of Failures in Microservices through Causal Discovery"
_NeurIPS.cc/2022/Conference — NeurIPS 2022 Accept_

### Official Review · Reviewer_TL8z · 2022-07-09

**Rating:** 4
**Confidence:** 3
**Soundness:** 3 good
**Presentation:** 3 good
**Contribution:** 3 good

**Summary:**

This paper proposes a causal discovery algorithm for quickly detecting the root cause of failure in microservice systems. The proposed algorithm treats a failure as an intervention on the root cause to learn the underlying causal structure. It only learns the portion of the causal graph related to the root cause, instead of relying on conditional independence tests. The authors compare their proposed solution with a modified version of the PC algorithm and some existing work on root cause analysis. The results show the effectiveness of the proposed algorithm (in terms of top-k precision) while keeping the execution time reasonable.

**Questions:**

. Did you compare the proposed approach with existing work using large-scale, real-world data?

. How does your method compare with [Ref-a] and [Ref-b] above?



**Limitations:**

No, the authors didn’t adequately address the limitations and potential negative societal impact of their work.

**Strengths And Weaknesses:**

Microservices become popular in recent years. This paper addressed an important problem of diagnosing failures of microservices systems. The proposed approach seems outperform the existing methods and is scalable.

The evaluation of the proposed approach is limited. The comparision with existing work is only conducted on a synthetic dataset. No real data is used in quantitative comparisons. It would be better to compare the proposed approach with existing work using large-scale, real-world data. The evaluation on Sock-shop is a small-scale test bed.  It consists of 13 microservices and only two failures (CPU hog and memory leak) are simulated. Therefore, the quantitative evaluation is rather limited.

Furthermore, there are many existing causal discovery-based methods for root cause analysis of failures in large-scale service systems.  For example, the following work also adopts a PC-based method for causal inference:

[Ref-a] Satoru Kobayashi, Kazuki Otomo, Kensuke Fukuda, and Hiroshi Esaki, Mining causality of network events in log data, in TNSM 2018.

The following work also adopts a FCI-based method for diagnosing failures of cloud service systems:

[Ref-b] Yujun Chen et al., Outage Prediction and Diagnosis for Cloud Service Systems, Proc. The Web Conference 2019, San Francisco, May 2019.

It would be better if the authors can discuss and compare with the above closely related work.

In the proposed Algorithm 1 (Root Cause Discovery Algorithm), it is not clear how F-Node is set. More details could be given here.

---

> ### Author Response · Authors · 2022-08-02
> **RCD relies on $\Psi$-PC to quickly find the root cause which is different than PC which only works with observational data**
>
> Thank you for reviewing our paper and for the suggestions to strengthen our work.
>
>
>
> > In the proposed Algorithm 1 (Root Cause Discovery Algorithm), it is not clear how F-Node is set. More details could be given here.
>
>
>
> The F-node is a tool proposed by Pearl to represent the observational and interventional distribution in a single causal graph [A]. It gives us the ability to identify the changes between the two distributions. To represent F-node in our dataset, we can assign any value to F-node if it is different between the two datasets. In our experiments, we used 0 for the failure-free dataset and 1 for the failure dataset. Thank you for pointing it out; we will add the clarification in Algorithm 1.
>
>
>
> > Did you compare the proposed approach with existing work using large-scale, real-world data?
>
>
>
> A few of the authors are in the process of obtaining access to real-world data of a large microservice-based application from a major software company. Although this access was not granted on time for publication, we expect to be able to run and include these experimental results in the camera-ready version.
>
>
>
> > How does your method compare with [Ref-a] and [Ref-b] above?
>
>
>
> Most solutions like [Ref-a] work in multiple steps such as collecting time-series data from system logs, running the PC algorithm to generate the causal graph, and finally ordering the nodes based on a random walk algorithm or the most common edges between multiple causal graphs. However, all these steps can introduce errors that will cause a high false positive rate in finding the root cause of the failure. In this case, the effectiveness of [Ref-a] is as good as the quality of the logs collected from the systems that can be poles apart for the two different modules of the same system. For instance, consider two services of a system where one service has extensive logging capabilities which the other lacks. In this case, the algorithm will not be able to detect the second service because it does not produce useful logs. Getting useful logs from the service will require application instrumentation. However, we do not require system logs; we only rely on metrics data (such as CPU and memory utilization) for services that are already available for all services and are easy to collect.
>
>
>
> Moreover, PC being an observational method, cannot identify the root cause and many parts of the full causal graph. Therefore, any method that uses PC to build the causal graph to identify the root cause needs to have a post-processing step to find the root cause given the causal graph. However, our proposed solution uses both observational and interventional data to pinpoint the root cause in the “one-shot” fashion.
>
>
>
> We believe [Ref-b] tries to solve a different problem – predicting the outages from alert signals. Their secondary goal is to help developers diagnose the problem by localizing it to a few services. To that extent, they construct an undirected graph using FCI to build the relationship between the alerts and outages. Later, they used this causal graph for outage diagnosis as well. However, the skeleton (undirected causal graph) for both PC and FCI is the same. Hence, for finding the root cause of failure in our setup, their solution is like $\Psi$-PC. Similar to $\Psi$-PC, it will incur a long runtime when the number of nodes increases because it relies on vanilla FCI.
>
>
>
> [A] Judea Pearl. Causal diagrams for empirical research. Biometrika, 82(4):669–688, 1995.

---

### Official Review · Reviewer_Y723 · 2022-07-12

**Rating:** 4
**Confidence:** 2
**Soundness:** 2 fair
**Presentation:** 2 fair
**Contribution:** 2 fair

**Summary:**

This paper formulates the root cause analysis problem as a causal inference task by modeling fault as an intervention on the failing node. A Root Cause Discovery algorithm inspired by existing FCI algorithm for detecting interventional targets is proposed. Additional node called F-node is introduced to represent the effect of the intervention. The proposed approach relies on the PC algorithm to learn causal graphs. It   reduces computation overhead of conditional independence tests by using a hierarchical approach and a localized learning algorithm that focuses on only learning the neighborhood of the F-NODE. Evaluation of this algorithm with synthetic and real-world dataset shows that the proposed approach generally outperforms the baseline methods.



**Questions:**

- How is the value of subset size determined?
- How significant are the false negative rates and/or recall numbers?
- A recent work [2] published in KDD’22 also models failures as interventions in their proposed approach. Could you briefly provide insights on additional benefits of the proposed approach?

[2]  Causal Inference-Based Root Cause Analysis for Online Service Systems with Intervention Recognition: https://arxiv.org/pdf/2206.05871.pdf


**Limitations:**

The proposed approach ignores the existence of latent variables. Also, the algorithm cannot process time-series data. These limitations are mentioned in the paper as future works.

There is no societal impact of this work that needs to be addressed.

**Strengths And Weaknesses:**

Strengths

- The proposed algorithm addresses a challenging problem: quick and accurate root cause diagnosis of failures in complex microservice systems. The localized learning for causal discovery approach is a good fit for this use case.

- Experimental evaluation on synthetic and test dataset show improvement in runtime and detection accuracy compared to the baseline approaches. A real-world failure is also diagnosed using the proposed approach to demonstrate the algorithm's superiority over the baseline methods.

Weaknesses

- Although the addition of a hierarchical approach reduces overall execution time with reduced conditional independence tests, it is not clear how to select the subset size or how the subset size impacts the accuracy of the algorithm. Experimental results on different subset sizes would provide useful insights.

- The evaluation metrics include precision numbers but no measurement of false negatives are presented.

- Inclusion of SOTA ML based techniques like Sage[1] in the comparative study would strengthen the overall evaluation of the proposed approach.

[1] Yu Gan, Mingyu Liang, Sundar Dev, David Lo, and Christina Delimitrou. 2021. Sage: Practical and Scalable ML-Driven Performance Debugging in Microservices. In ASPLOS. 135–151.

---

> ### Author Response · Authors · 2022-08-02
> **RCD differs from other works because we use observational and interventional data, we do not make any parametric assumptions nor require any domain knowledge**
>
> We appreciate your helpful comments!
>
> > […] Experimental results on different subset sizes would provide useful insights.
>
> In all our experiments in the main paper, we use a fixed value for subset size ($\gamma$) = 5. Because of the page limit, we could not include the sensitivity analysis of the subset size in the main paper. However, we discuss the effect of the subset size on the accuracy of the algorithm in the supplementary section (L707).
>
> > How significant are the false negative rates and/or recall numbers?
>
> The most common metric to measure the effectiveness of an algorithm to find the root cause is top-k precision [A, B].  In our simulations we only had one root cause for simplicity. Recall is TP/(TP+FN), i.e., correctly identified root causes divided by all root-causes. With a single root-cause, this metric is either 1 or 0 and is the same as top-k precision. We will add simulations with multiple root-causes in the camera-ready since the method is able to handle this case.
>
> > Inclusion of SOTA ML based techniques like Sage [1] in the comparative study would strengthen the overall evaluation of the proposed approach.
>
> Our setting differs from Sage in a few ways. 1) Sage considers unlabeled data, whereas, in real-world applications, the normal and anomalous datasets are labeled based on some static rules (for example rules for alerts) or with the help of an anomaly detection system. 2) We consider interventional data coupled with observational whereas Sage only works with observational data that can lead to incomplete causal graphs (with observational data, we might only be able to find the correlation between two variables but with interventional data we can infer the cause and effect). 3) Sage relies on a set of static rules that come from expert knowledge and will be different for other applications. On the flip side, we do not consider any such static rules while building the causal graph that makes our solution more applicable. 4) Finally, as pointed out in the paper itself, one of the most important assumptions is that Sage can only detect the root cause of a failure if it has observed a similar failure in the past. However, RCD does not need to make any such assumptions. In any case, we plan to compare the proposed technique with Sage in the camera-ready version.
>
> > A recent work [2] published in KDD’22 also models failures as interventions in their proposed approach. Could you briefly provide insights on additional benefits of the proposed approach?
>
> Thank you very much for pointing out this very relevant independent work. We will surely discuss and cite this related work in the camera-ready version.
>
> Like Sage, the major dissimilarity we have with this work is that we do not rely on any domain knowledge to build the causal graph. With the call graph given, KDD’22 does not need to estimate the causal structure and hence, the problem reduces to inferring root cause from the call graph. On the flip side, we assume no knowledge of causal/communication structure which makes our solution more general and applicable. Moreover, KDD’22 uses a set of static rules to build a causal graph from the call graph. These rules are domain-specific and therefore do not work for all applications. For instance, one of the rules is that the callee’s traffic load affects the caller’s latency. This statement holds only when the communication between two services is a blocking call. If we have a non-blocking call where the caller does not wait for the response from the callee, this statement will lead to the wrong causal graph.
>
> Another key difference is that they use the interventional dataset to build a regression model that they later use to assign an anomaly score to all the services. However, as stated earlier, regression models require parametric assumptions that might not hold under certain settings.  However, we use the interventional dataset to find the root cause without making any parametric assumptions or relying on the data about past failures. Finally, their solution tries to find the most ancestral node where the failure can be observed (Algorithm 2). This will lead to low accuracy if the failing node has many ancestors.
>
> In conclusion, our approach is more applicable because it does not rely on any domain knowledge or require any historical data about past failures or make any parametric assumptions. Hence our solution for root cause detection problem can be extended to any domain where a failure can be modelled as an intervention.
>
> [A] Ma, M., Xu, J., Wang, Y., Chen, P., Zhang, Z., & Wang, P. (2020, April). Automap: Diagnose your microservice-based web applications automatically. In Proceedings of The Web Conference 2020 (pp. 246-258).
>
> [B] Wu, L., Tordsson, J., Bogatinovski, J., Elmroth, E., & Kao, O. (2021, May). MicroDiag: Fine-grained Performance Diagnosis for Microservice Systems. In 2021 IEEE/ACM International Workshop on Cloud Intelligence (CloudIntelligence) (pp. 31-36). IEEE.

---

> > ### Comment · Reviewer_Y723 · 2022-08-09
> > **Clarification on the responses above.**
> >
> > Thanks for the detailed responses.
> >
> > About false negatives: I think more real world / synthetic cases could be considered to get the false negative aspect of the algorithm, which would also help in comparing with other relevant approaches (e.g., Sage) that have different setting but the same goal as the proposed work.
> >
> > Related works and contributions: Thanks for adding a comparative discussion on the KDD'22 work in your response. Not relying on domain knowledge could be a useful feature but I think that feature is only useful if the proposed approach shows no performance tradeoff for avoiding the need of such rules. The core idea of the proposed work, modeling failure as an intervention on the root cause, is very similar to the KDD'22 work. The variation introduced in this work for avoiding dependence on domain knowledge may have positive/negative impact compared to KDD'22. I think the contribution of this paper would be more clearer with additional experiments that highlights these impacts.

---

### Official Review · Reviewer_mejF · 2022-07-14

**Rating:** 4
**Confidence:** 4
**Soundness:** 2 fair
**Presentation:** 2 fair
**Contribution:** 2 fair

**Summary:**

Due to the complex and dynamic inter-module correlation structure, the microservice system in the cloud-native environment produces complicated failure behaviors. To efficiently diagnose the fault and identify the root cause, this study draws on the idea of the $\psi$-FCI algorithm, treating faults as interventions on root causes to learn the underlying causal structure. With this, the authors design a causal graph construction algorithm, called $\psi$-PC, which degenerates from the $\psi$-FCI algorithm based on the assumption of no confounding variables, randomly grouping all nodes, and using a two-step hierarchical test to optimize computational efficiency. Finally, the generated and real data are used to verify the algorithm's advantages in terms of diagnostic accuracy and computational efficiency.

**Questions:**

The abstract fails to highlight the problems of existing research and the essential difference between this method and previous studies. Likewise, while Section 2 comprehensively covers the relevant concepts, it does not explain the inadequacies of previous studies and what is the specific observation / motivation of this study.

The authors mention in Section 3: “The key benefit of modeling the failure as an intervention is that it enables us to translate the problem of finding the root cause of failure into finding the interventional target. In addition, this view allows us to leverage the recent developments in the causal discovery literature that can identify intervention targets. ” I disagree that this is a convincing statement, as the authors only say the benefits but do not justify the validity of underlying assumptions.

Furthermore, the paper lacks real-world case studies to aid in understanding the algorithmic processes and details. For example, how to identify the intervention target in $\psi$-PC, and what is the relationship between the intervention target and the root cause. It is recommended to use real-world running examples and causal e-and-effect graphs in real diagnostic scenarios to illustrate which nodes are (or could be) interventions and which are intervention targets? Figure 1 does not help to evaluate the key observations’ validity / reasonableness in microservices troubleshooting scenarios. Besides, the author mentions an assumption (observations) without any explanation: “a fault changes the generative mechanism of the failing node.” I would like to see why this assumption or observation holds true with examples.

In the first phase of $\psi$-PC, is it a random division? Or should refer to any prior knowledge? How to ensure that the division place no adverse effect on the result? Since it also divides the causal structure into local graphs, how to theoretically ensure that the local graph will always contain the correct root cause compared to the global graph.

The section analyzing the algorithmic complexity mentions: " By decoupling learning the causal graph from finding the interventional targets, we can improve runtime by significantly reducing the number of CI tests." However, even if we use an iterative PC algorithm, the complexity can be controlled by limiting the depth of CI tests. In addition, adopting the heuristic search algorithm can also generate more accurate results on specious causal graphs. In other words, performing the CI tests thoroughly and iteratively is unnecessary. Therefore, it is recommended to discuss and demonstrate the computational complexity by comparing with various optimization baselines (such as $\psi$-FCI, PC, or other related methods such as PCMCI and Granger's test, etc.).

Lastly, I noticed that the authors compared ϵ-Diagnosis and AutoMAP using synthetic data. The diagnostic goals for these baselines appear to be different from $\psi$-PC. They aim to find the deepest nodes in the failure propagation chain and consider whether they are highly correlated with failure patterns. This idea stems from engineers' intuition in manual troubleshooting. It is suggested that the author analyze the adaptability of different diagnostic ideas and simulation methods.

**Limitations:**

Not applicable.

**Strengths And Weaknesses:**

Strengths:
- The problem has real research significance and economic value, especially for large-scale cloud system failure operation and maintenance
- The efficiency advantage of the proposed solution makes it more adaptable to real system scenarios

Weaknesses:
- Lack of clarification on the primary observation/motivation of the paper
- Running cases are not provided to show many details and results of the proposed method
- Insufficient theoretical contribution as $\psi$-PC is a simplification on the basis of $\psi$-FCI

---

> ### Author Response · Authors · 2022-08-02
> **A fault changes the distribution of the failing node allowing us to translate the problem of finding the root cause into interventional target identification.**
>
> Thank you for your detailed and constructive feedback! We apologize for not being clear and will try to address your concerns in the following;
>
> > […] the authors design a causal graph construction algorithm, called $\Psi$-PC, which degenerates from the $\Psi$-FCI algorithm
>
> $\Psi$-PC is indeed a special case of $\Psi$-FCI that works under relaxed constraints. However, our key contribution is not $\Psi$-PC but RCD that uses $\Psi$-PC as a tool to quickly find the interventional target. In our experiments, we compared our solution with $\Psi$-PC and showed the gain of using RCD to find the root cause. Moreover, we also provide theoretical proof that states RCD can find the correct root cause under loose assumptions (causal sufficiency and extended faithfulness).
>
> > The abstract fails to highlight the problems of existing research and the essential difference between this method and previous studies
>
> Most existing algorithms only work with an observational dataset, and not observational _and_ interventional data as we use. This limits their ability to learn the underlying causal structure under different states of the system. Furthermore, they make parametric assumptions to simplify the problem and only work under strict constraints (such as only with linear relations between the metrics). Our proposed solution does not make any such assumptions and therefore is more applicable. Thank you for pointing this out; we will add these clarifications to the main paper.
>
> > The authors mention in Section 3: “The key benefit of modeling failure as an intervention is that it enables us to translate the problem of finding the root cause of failure into finding the interventional target. […]” I disagree that this is a convincing statement, as the authors only say the benefits but do not justify the validity of underlying assumptions. [...] the author mentions an assumption (observations) without any explanation: “a fault changes the generative mechanism of the failing node.” I would like to see why this assumption or observation holds true with examples. […] how to identify the intervention target in $\Psi$-PC, and what is the relationship between the intervention target and the root cause. […] which nodes are (or could be) interventions and which are intervention targets?
>
> s we mention in section 2 (L 79), an intervention is the process of changing the generative mechanism of a random variable. According to Pearl's causal framework, an intervention on a variable is assumed to only affect its causal generative mechanism, which is known as the modularity assumption [A].
>
> For example, consider two microservices A -> B where for every request A receives, it performs some processing and forwards the response to B. In other words, if the latency of A increases, the latency of B will also grow. Further consider that during the normal state of the system, mean latency of A is 50 ms which increases to 100 ms because of a bug in the system during the anomalous state. From this example, we can observe that the failure changes the distribution of the node where error occurred (A). The increase in the latency of A will cause the latency of B to increase as well. i.e., the distribution of B will also get effected. However, the root cause of the error is still A and B is a node on the error propagation chain. We can identify this because the distribution of B given A does not change and therefore, we can remove B from the list of candidate root causes.
>
> Continuing the previous example, if we consider the latency of A and B to be the nodes in a causal graph with a directed edge from A to B, we will consider the failure an intervention on A, or the interventional target is A. Given a causal graph and a pair of observational and interventional datasets, $\Psi$-PC finds the interventional target which in our case is the root cause of the failure.
>
> More details about interventions and mapping the problem of pinpointing the root cause to finding the interventional target can be found in Sections 2 and 3 of our paper. Thank you for your question; we will add this clarifying text in the camera-ready version.
>
> [A] Judea Pearl. Causality. Cambridge university press, 2009

---

> > ### Comment · Reviewer_mejF · 2022-08-08
> > **Response to Paper12457 Authors**
> >
> > Thanks to the author for such a thorough reply explaining many details of the RCD method. But I am still concerned about the reasonableness of modeling faults as interventions, especially whether it matches the actual situation in operation and maintenance scenarios in large-scale production systems. The paper still lacks real operating cases to justify the motivation for designing RCD, leading to the soundness of the paper requiring a deep revision. Given the work is instructive for cloud operation and maintenance, I have raised the score to the borderline to see if it has a chance of being accepted.

---

> > > ### Author Response · Authors · 2022-08-09
> > > **Applicability of modeling failure as an intervention**
> > >
> > > Thank you very much for your feedback and for revising the score.
> > >
> > > As we mentioned earlier in our response, the failure changes the distribution of the faulty node, which allows us to consider the failure as an intervention on the failing node. We also provide an example that shows the intuition behind this assumption. To demonstrate the applicability of this assumption in real-world applications, we have applied RCD to two case studies (Section 5) – with simulated failures (Sock-shop) and on a real-world service failure dataset. We are also in the process of obtaining a dataset about the failures in a cloud-based application from a big software company. We hope to be able to run RCD and include the experiment results in the camera-ready.
> > >
> > > Furthermore, Reviewer Y723 pointed us to a recent independent study [A] published in KDD'22 that also models the failure as an intervention on the faulty node. However, they make parametric assumptions and cannot find the root cause without domain knowledge. The detailed comparison between Y723 and [A] is in response to Reviewer Y723's comments.
> > >
> > > [A] Li, Mingjie, Zeyan Li, Kanglin Yin, Xiaohui Nie, Wenchi Zhang, Kaixin Sui, and Dan Pei. "Causal Inference-Based Root Cause Analysis for Online Service Systems with Intervention Recognition." arXiv preprint arXiv:2206.05871 (2022).

---

> ### Author Response · Authors · 2022-08-02
> **RCD differs from other works because it uses observational and interventional data and does not make any parametric assumptions.**
>
> > How to ensure that the division places no adverse effect on the result? Since it also divides the causal structure into local graphs, how to theoretically ensure that the local graph will always contain the correct root cause compared to the global graph?
>
> The commonly adopted faithfulness assumption from causal inference extends to the interventional setting [E]. In our case, it ensures that no matter how we split the data, the root cause will always be adjacent to the F-node. Dividing the data into small segments only helps to reduce the number of CI tests to find the immediate neighbors of the F-node. The only side effect of the division is that it might lead to false positives because we could not condition on the variables that were removed in the first phase but still no root cause will be missed. We further argue that false positives are tolerable for our application if we can find the actual root cause within reasonable time (say, order of a minute or even a few minutes). We also provide the theoretical proof in supplementary section (L 605) showing that the RCD will always find the root cause under the given assumptions.
>
> > […] it is recommended to discuss and demonstrate the computational complexity by comparing with various optimization baselines (such as $\Psi$-FCI, PC, or other related methods e.g. PCMCI and Granger's test, etc.
>
> PC, PCMCI, and most heuristic-based search algorithms only work with an observational dataset, and not observational and interventional data as we use. This limits their ability to learn the underlying causal structure. Relying only on observational data limits their ability to find the causal structure related to the root cause. However, our method is capable of learning that causal structure by systematically leveraging the distributional invariance not only within the observational data, but also across observational and interventional dataset. This has been proven to be strictly more informative than the former [F, G].
>
> Furthermore, they make parametric assumptions to simplify the problem and only work under strict constraints (such as only with linear relations between these metrics). However, these assumptions do not hold in real-world applications. For instance, consider a case where the latency of a microservice grows linearly with respect to the CPU utilization. However, after a point, the degree of parallelism exceeds the gain of running multiple requests in parallel (because of the increasing cost of context switches). After which the latency increases exponentially with respect to the number of incoming requests. This non-linearity has been pointed out myriad times in the systems literature [A, B, C]. To capture such common behavior, most existing works make parametric assumptions.
>
> $\Psi$-FCI is the exception as it considers multiple interventional datasets with the presence of latent variables. Note that $\Psi$-FCI and $\Psi$-PC are equivalent under our assumption that there are not any latent confounders. We provide comparison with $\Psi$-FCI both qualitatively and experimentally.
>
> Other work such as Sieve [D] uses Granger Causality test to decide whether there should be an edge between two variables and later use the constructed graph to infer the root cause. This approach leaves out edges which could have been removed by conditioning on other variables. In addition, a Granger Causality based approach assumes a parametric functional form, usually a linear model, for the test. As noted in response to question 3 above, such parametric assumptions are not always valid for modeling dependencies between systems’ metrics.
>
> [A] Zhao, et al. "Rhythm: component-distinguishable workload deployment in datacenters." In Proceedings of the Fifteenth European Conference on Computer Systems, pp. 1-17. 2020.
>
> [B] Masouros, et al. "Rusty: Runtime interference-aware predictive monitoring for modern multi-tenant systems." IEEE Transactions on Parallel and Distributed Systems 32, no. 1 (2020): 184-198.
>
> [C] Barve, et al.. "FECBench: A holistic interference-aware approach for application performance modeling." In 2019 IEEE International Conference on Cloud Engineering (IC2E), pp. 211-221. IEEE, 2019.
>
> [D] Thalheim, et al. "Sieve: Actionable insights from monitored metrics in distributed systems." In Proceedings of the 18th ACM/IFIP/USENIX Middleware Conference, pp. 14-27. 2017.
>
> [E] Judea Pearl. Causality. Cambridge university press, 2009
>
> [F] Karren, et al. Characterizing and learning equivalence classes of causal dags under interventions. In International Conference on Machine Learning, pages 5541–5550. PMLR, 2018.
>
> [G] Amin, et al. Causal discovery from soft interventions with unknown targets: Characterization and learning. Advances in neural information processing systems, 33:9551–9561, 2020

---

> ### Author Response · Authors · 2022-08-02
> **RCD finds the root cause in a "one-shot" fashion and therefore does not require any post-processing step. The developer's intuition can be used to improve the performance of RCD further.**
>
> > In the first phase of $\Psi$-PC, is it a random division? Or should refer to any prior knowledge?
>
> In our experiments, we randomly split the dataset into subsets of size $\gamma$. However, we believe that the prior/expert knowledge can be used to reduce the number of CI tests further. For instance, packing highly correlated metrics in the same subset is better because we need to condition on all of them together to remove them from the candidate list and to find the most ancestral node in the subset where the effect of the failure can be observed. However, if we put these metrics in different bins, they will survive to the next stage and hence will increase the number of CI tests.
>
> > Lastly, I noticed that the authors compared $\epsilon$-Diagnosis and AutoMAP using synthetic data. […] They aim to find the deepest nodes in the failure propagation chain and consider whether they are highly correlated with failure patterns. This idea stems from engineers' intuition in manual troubleshooting. It is suggested that the author analyze the adaptability of different diagnostic ideas and simulation methods.
>
> We argue that these baselines are relevant as they also try to find the root cause of failure. In our simulation, we generate graphs such that the failing node and the node where the effect is observed are far apart. The key point where our solution differs is that our solution does not require any post processing step such as finding the deepest node. RCD outputs a causal graph where the immediate neighbors of F-node are the actual root cause. Please note that we acknowledge that if engineers’ intuition is used to fine tune our algorithm, that could get even better results. Such as packing relevant metrics into the same subset or removing some of the normal services from the data.

---

### Public Comment · ~Jonas_M._Kübler1 · 2023-01-11
**Availability of Code & Data**

Dear Authors,
In the paper checklist, you mention that you "plan to share [y]our code and the data in the future".
Did you already make them available somewhere?

Best,
Jonas

---

> ### Public Comment · ~Azam_Ikram1 · 2023-01-12
> **Code availability**
>
> Dear Jonas,
>
> Thank you for your interest in our paper!
>
> We are currently in the process of patenting the proposed idea and therefore might not be able to share the code yet. I will put a link to our github repository in the main paper and will populate it as soon as the patenting is done.
>
> Best,
> Azam

---

> ### Public Comment · ~Azam_Ikram1 · 2023-01-27
> **Code is available on our github repo**
>
> Dear Jonas,
>
> I just wanted to let you know that we have made our code publicly available on our [Github repository](https://github.com/azamikram/rcd). It also includes instructions on how to set up the environment and run the code.
>
> For any further information or query, please feel free to reach out to me at mikram@purdue.edu.
>
> Happy coding!
>
> Best,
> Azam

---

### Meta-Review · Area_Chair_dN4a · 2022-08-26

**Recommendation:** Accept
**Confidence:** Less certain

**Metareview:**

The paper provides a new approach for learning root causes of failures and is targeted to micro-services. One of the main shortcomings of the paper raised by reviewers was in the evaluation. On the one hand, reviewers pointed out a number of very relevant related works that are not compared with this work, and on the other hand, the bulk of the quantitative assessments are done using synthetic data, and there were some questions about the level of realism of the Sock-store experiment. That said, the rebuttal had a good analysis of the relationship with those related works and argued for why they are not entirely comparable. I do think the evaluation could be stronger; in fact, I think the paper could have more impact and more visibility in a systems conference, but that would require a more complete evaluation on more real systems (it is instructive to compare the evaluation of this paper with that of the Sage paper brought up by one of the reviewers). However, I think the combination of an interesting algorithmic contribution, strong results on simulated data and a compelling real-world case study puts this paper above the bar.

**Award:**

No

---

### Decision · Program_Chairs · 2022-09-14

Accept